# Prevalence and predictors of COVID-19 vaccination hesitancy among healthcare workers in Sub-Saharan Africa: A systematic review and meta-analysis

Eustes Kigongo[1]*, Amir Kabunga[2], Raymond Tumwesigye[3], Marvin Musinguzi[4], Ronald Izaruku[5], Walter Acup[4]

1 Faculty of Public Health, Department of Environmental Health and Disease Control, Lira University, Lira, Uganda, 2 Faculty of Medicine, Department of Psychiatry, Lira University, Lira, Uganda, 3 Faculty of Nursing and Midwifery, Department of Nursing, Lira University, Lira, Uganda, 4 Faculty of Public Health, Department of Community Health, Lira University, Lira, Uganda, 5 Department of Library and Information Services, Lira University, Lira, Uganda

* ekigongo@lirauni.ac.ug

## Abstract

### Background

The COVID-19 vaccination is regarded as an effective intervention for controlling the pandemic. However, COVID-19 vaccine hesitancy is hampering efforts geared towards reducing the burden of the pandemic. Therefore, examining COVID-19 hesitancy and its predictors among healthcare workers is essential to improving COVID-19 uptake. In sub-Saharan Africa, the pooled proportion of COVID-19 vaccine hesitancy is yet to be known.

### Purpose

The present study was to estimate the pooled proportion of COVID-19 vaccine hesitancy and its predictors among healthcare workers in Sub-Saharan Africa.

### Methods

A systematic search of articles was conducted in PubMed, Science Direct, African Journal Online, and Google Scholar. Data was extracted with the help of Excel. Data analysis was conducted using STATA 17. Heterogeneity in the studies was assessed using Cochrane Q and $I^2$ tests. A random effects model was used to examine the pooled estimates to determine if heterogeneity was exhibited.

### Results

A total of 15 studies involving 7498 participants were included in the final analysis. The pooled prevalence of COVID-19 vaccination hesitancy among healthcare workers was 46%, 95% CI (0.38–0.54). The predictors of COVID-19 hesitancy were negative beliefs towards vaccine 14.0% (OR = 1.05, 95% CI: 1.04, 1.06), perceived low risk of COVID-19

**Data Availability Statement:** All data are fully available without restriction and have been attached as Supplementary 3.

**Funding:** The authors received no specific funding for this work.

**Competing interests:** The authors have declared that no competing interests exist.

infection 24.0% (OR = 1.25, 95% CI: 1.23, 1.28), and vaccine side effects 25.0% (OR = 1.23, 95% CI: 1.21, 1.24).

## Conclusion

The data revealed generally high hesitancy of COVID-19 vaccine among health workers in Sub-Saharan Africa. Future COVID-19 adoption and uptake should be improved by national and individual level efforts. In Sub-Saharan Africa, it is crucial to address the myths and obstacles preventing healthcare professionals from accepting the COVID-19 vaccination as soon as feasible since their willingness to get the vaccine serves as an important example for the broader public.

## Introduction

Since it was initially discovered in Wuhan, China, in late December 2019, coronavirus disease-2019 (COVID-19), a highly contagious illness brought on by the SARS-CoV-2 virus, has become a source of concern for the general public worldwide [1]. On February 12, 2020, the World Health Organization (WHO) formally suggested calling this infectious disease coronavirus disease 2019 (COVID-19), and on March 11, 2020, COVID-19 was determined to have the characteristics of a worldwide pandemic [2]. While vaccination is the best way to prevent the spread of COVID-19, with the introduction of COVID-19 vaccines, the general resistance to and refusal to receive vaccinations appears to have increased, and what is most concerning is that this behavior is being noticed among healthcare workers in the face of a pandemic [3]. For example, in a total sample of 76,471 HCWs from 21 countries, the average prevalence of COVID-19 vaccine hesitancy was 22.51%, ranging from 4.3 to 72% [4].

Similar to other RNA viruses, SARS-CoV-2 is a member of the b-coronavirus subgenus. During host adaptation, SARS-CoV-2 experiences a high level of genomic mutation, which presents a considerable challenge to current therapeutic approaches and preventative measures [5]. Medical professionals and researchers from all around the world have been looking for effective treatment options, such as antiviral medicines, immunotherapy, and vaccines, to better prevent novel coronavirus infections and stop the outbreak [5]. However, due to the lack of treatment for COVID-19, the failure of viral infection and vaccine-induced immunity to stop the spread of the epidemic, and the emergence of antigenically different variations, herd immunity against SARS-COV-2 cannot yet be obtained [6]. A safe and effective vaccine is the most effective and dependable way to build up the population's immune system (herd immunity) to prevent recurrent infections [5].

SARS-CoV-2 vaccinations are safe and effective in lowering SARS-CoV-2-related deaths, symptomatic cases, severe cases, and infections globally [7]. Healthcare workers have easier access to populations of COVID-19 patients during routine diagnostic and treatment activities and are at much greater risk of contracting COVID-19 than other populations, so the Advisory Committee on Immunization Practices (ACIP) proposed prioritizing healthcare workers' vaccination in December 2020 [8]. There have been reviews showing a moderate level of healthcare workers' acceptance of the COVID-19 vaccine [1]. According to Shui and colleagues, it is crucial to implement initiatives to increase healthcare workers' acceptance of and readiness to get the COVID-19 vaccine to stop the disease from spreading [1].

An increasing body of research indicates that COVID-19 vaccinations are both safe and efficacious [9]. COVID-19 vaccinations lower the risk of infection and serious consequences

[9]. COVID-19 vaccinations provide benefits that outweigh the risk of uncommon adverse effects [9]. Many COVID-19 vaccines were approved in different countries in Sub-Saharan Africa, including those from Janssen (Johnson & Johnson), Moderna, Pfizer/BioNTech, Sino-vac, and StraZeneca [10]. COVID-19 vaccinations have been recommended many public health organizations, including the WHO [9]. Despite the evident and confirmed benefits of COVID-19 vaccines, there is still vaccine hesitancy among many individuals in the world, including health care personnel. In Africa, the pooled COVID-19 vaccine acceptance rate was predicted to be 46% [11]. Vaccination-specific concerns (safety and effectiveness), a lack of evidence or information, antivaccine attitudes, and a lack of institutional trust are all reasons for vaccine hesitation [12]. Several studies have looked into the elements that contribute to COVID-19 vaccine hesitancy. Low educational attainment, ethnic disparities, rurality, and resistance to other vaccinations (e.g., influenza) are all common variables [12, 13]. However, research on the most important predictors of resistance to COVID-19 immunization is constantly emerging. While the general population's COVID-19 vaccination hesitancy rates have been well investigated and are reasonably well established, few studies have explicitly investigated COVID-19 vaccine hesitancy among health care personnel [13].

Healthcare workers are still on the front lines of the current pandemic, and countries have prioritized them as the first to be vaccinated [14]. Additionally, healthcare workers serve as a link between healthcare systems and patients. People in the community frequently rely on healthcare workers' knowledge and behaviors to guide their decisions about whether to accept or refuse the COVID-19 vaccine [15]. However, there has been an increase in reports of vaccination hesitancy [16]. Examining the level of acceptance of the COVID-19 vaccination and predictors of COVID-19 vaccination hesitancy among healthcare workers would thus assist policymakers, researchers, and health authorities in developing appropriate vaccine hesitancy interventions. However, among this group of people in the Sub-Saharan Africa region, there are currently no comprehensive reviews of COVID-19 hesitancy and its predictors among healthcare workers. This review aimed to provide a synthesis of evidence on prevalence and predictors of COVID-19 vaccination hesitancy among healthcare workers in Sub-Saharan Africa. In this regard, the review attempted to answer the following questions;

- What is the proportion of COVID-19 vaccination hesitancy among healthcare workers in Sub-Saharan Africa?

- What are the predictors of COVID-19 vaccination hesitancy among healthcare workers in Sub-Saharan Africa?

## Methods

This systematic review and meta-analysis were conducted in accordance with the guidelines of Preferred Reporting Items for Systematic Review and Meta-Analysis (PRISMA) to ensure a rigorous selection and reporting of studies. The study was registered in the Prospective Register of Systematic Review (PROSPERO) (ID: CRD42022359141).

### Search strategy

We searched the following electronic bibliographic databases: Google Scholar, African Journal Online, PubMed, and Science Direct, using a combination of keywords and appropriate thesauri for COVID-19 vaccination hesitancy and healthcare workers. The selected keywords were a vaccine, COVID-19 or SARS, hesitancy, refusal, acceptance, or acceptability. The keywords were conveniently selected based on the study objectives. PubMed was used to create the initial combination of search phrases that were customized for other databases using

appropriate thesauri. The literature search was supplemented with materials from institutional repositories as well as studies from reference lists of similar review articles. The detailed strategy has been attached (S1 Table).

## Inclusion criteria and exclusion criteria

All studies conducted in English about COVID-19 vaccination hesitancy among health workers in Sub-Saharan Africa since December 2020 were considered. Our inclusion criteria included only studies reporting on predictors of COVID-19 vaccination hesitancy in English, primary studies, all the types of vaccines approved for use by the FAO, and quantitative studies published from December 2020 to October 2022 within sub-Saharan Africa (SSA). We excluded COVID-19 studies reporting animal studies, reviews, commentaries, letters to the editors, studies in which raw data could not be transformed, and studies on hesitancy rates explicitly referring to participants other than healthcare workers in Sub-Saharan Africa.

## Study and data management

After searching and collecting articles, duplicate files were removed. The screening was done at two levels: the title and abstracts, and then the full-text screening. This was performed by two independent reviewers to assess their potential relevance for full review using the EndNote software, done against the set inclusion and exclusion criteria. Any discrepancies were resolved by discussion and consulting the third reviewer. Electronic records of the included and excluded studies were kept for audit purposes, specifying reasons for any exclusion.

## Data extraction

The data were extracted using a Microsoft Excel template. The form was tested and revised iteratively as needed. The following were included: The first name of the first author, year of publication, study title, country, and study design, period of conducting a survey, study participants, sample size, response rate, and proportion of males, average age, reported hesitancy, and predictors of hesitancy (S2 Table).

## Quality assessment for the included papers

The risk of bias was assessed using the validated quality assessment checklist for prevalence studies from Hoy and colleagues [17]. The tool has nine items that generate a total score of 9 (S1 Raw data). These items include the target population, sampling frame, sampling, response rate, data collection, study case definition, study instruments, and parameters for the numerator and denominator. The tools classify studies as low-risk (0–3), moderate-risk (4–6), and high-risk (7–9). Each study was assessed independently, and the majority of the studies were rated as having a low risk of bias. Studies with high risk of bias were not included in the final analysis.

## Ethical considerations

The study conducted did not require ethical approval because it utilized secondary data from previously published literature.

## Data synthesis and analysis

The data from individual articles was extracted using Microsoft Excel 2013 and exported to STATA (version 17; StataCorp, TX, USA) for all analyses. To determine the pooled prevalence of COVID-19 vaccination hesitancy, analysis was conducted on 15 individual studies, with the

main findings described in a table with narratives in texts. Due to the high heterogeneity (I2 = 91.96%, p≤0.001) among the included studies, a meta-analysis using the random effects model was performed to estimate the pooled prevalence of COVID-19 vaccination hesitancy among healthcare workers in sub-Saharan Africa. This was performed at a 95% confidence interval, and the results were presented using a forest plot. The Cochran's Q test was used to test heterogeneity using the I2 statistics, which were interpreted as low heterogeneity (25%), moderate heterogeneity (25%–50%), and high heterogeneity (>50%) [18]. We also estimated pooled odds ratios for factors associated with vaccine hesitancy, and statistical significance was determined at P<0.05 of the I2. An explanatory variable was eligible for inclusion if there was data from at least two of the included studies. Publication bias was also assessed using visual inspection of funnel plot asymmetry and a weighted Egger's regression test with P = 0.05 as a cutoff point to declare the presence of publication bias. Subgroup analysis was performed based on the potential sources of heterogeneity. Leave one out sensitivity analysis was also performed to assess the influence of single studies on the overall effect [19].

## Results

The electronic search yielded 13,647 articles, and 11,900 articles remained after the duplicates were deleted. After screening abstracts and titles, 35 articles remained and were subjected to full-text screening. Twenty articles were removed for different reasons: the full outcome of interest was not reported (n = 15) and the full text was not available (n = 5). The results are shown in Fig 1.

### Characteristics of the studies

Out of the 15 included studies, 1 was conducted in the Democratic Republic of the Congo (DRC), 1 from Ghana, 3 from Nigeria, 1 from Sierra Leone, 1 from Togo, 1 from Uganda, 7 from Ethiopia, and 1 from Tanzania. The sample size ranged from 108 to 811 participants, giving a total sample size of 7498 participants. The participants included doctors, medical laboratory scientists, allied health professionals, pharmacists, nurses, eye health workers, and medical students. The first survey was conducted between November and December 2020, and the most recent was between September and October 2022. The studies were all cross-sectional and published between 2020 and 2021. The majority of the included studies had a low risk of bias. The findings are summarized in Table 1.

### Pooled COVID-19 vaccine hesitancy level in Sub-Saharan Africa

The pooled prevalence of COVID-19 vaccination hesitancy rate in Fig 2 among healthcare workers was 46%, 95% CI (0.38–0.54). In this meta-analysis, a random effects model was executed as high heterogeneity (I$^2$ = 91.96%, p≤0.001) was detected within the included studies.

### Publication bias assessment

Fig 3 indicates funnel plot symmetry. The Eggers test (0.3256) also showed absence of potential publication bias.

### Sub-group and sensitivity analysis

Sub-regions were used for sub-group analysis, as indicated in Fig 4 (Central Africa, East Africa, and West Africa). The pooled estimated COVID-19 hesitancy rate in Central Africa was 72% (95% CI: 64%–80%), in West Africa it was 52% (95 CI: 43%–61%), and in East Africa it was 40% (95% CI: 30%–50%). Since we had more than 10 studies, leave one out sensitivity analysis

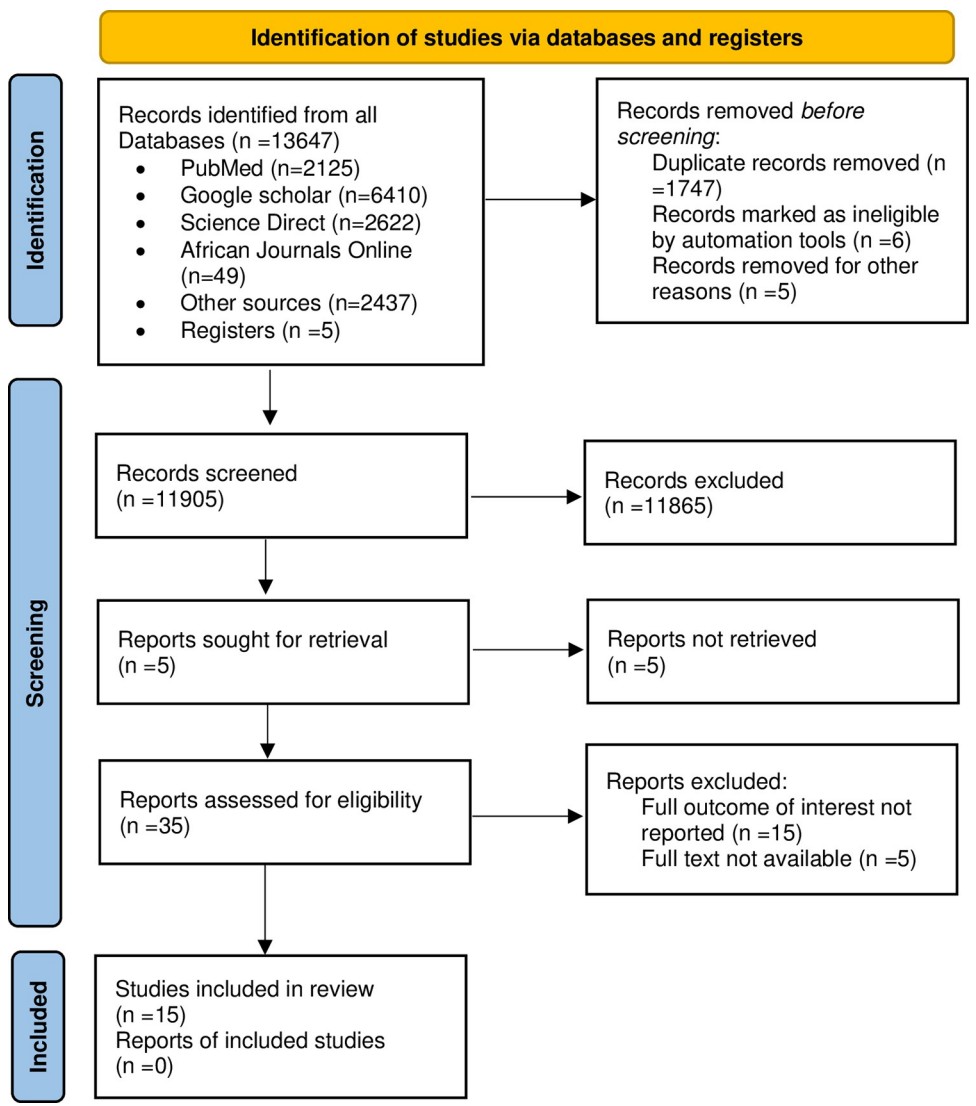

**Fig 1. The Preferred Reporting Items for Systematic Review and Meta-Analysis (PRISMA) flow diagram of the study selection process.**

was conducted to assess the impact of independent studies on the overall pooled effect. The results revealed that no single study had a significant impact on the overall prevalence and ranged from 44% (95% CI: 36% - 52%) to 48% (95% CI: 41% to 56%) (S1 Fig).

### The predictors of COVID-19 hesitancy among healthcare workers

Results in Table 2 show that the predictors of COVID-19 hesitancy among healthcare workers were negative beliefs towards vaccine 14.0% (OR = 1.05, 95% CI: 1.04, 1.06), perceived low risk of COVID-19 infection 24.0% (OR = 1.25, 95% CI: 1.23, 1.28), and vaccine side effects 25.0% (OR = 1.23, 95% CI: 1.21, 1.24).

## Discussion

This review investigated healthcare workers' pooled level of hesitation over the COVID-19 vaccine and related factors. Acceptance of the COVID-19 vaccination is a serious global

**Table 1. Characteristics of included studies.**

| SN | Author | Year | Country | Participants | Sample | Sampling technique | Period of Study | Males (%) | Average Age (%) | Hesitancy (n) |
|---|---|---|---|---|---|---|---|---|---|---|
| 1 | Abay et al [20] | 2022 | Ethiopia | CP | 464 | Convenience | Apr-May, 2021 | 59.2 | 20-29(51.8) | 72 |
| 2 | Adane et al [21] | 2022 | Ethiopia | HCW | 404 | Simple random | May 2021 | 50.5 | 20-30(40.1) | 141 |
| 3 | Aemro et al [22] | 2021 | Ethiopia | HCW | 440 | Simple random | May-Jun, 2021 | 62.4 | 26-30(56.0) | 192 |
| 4 | Amuzie et al [23] | 2021 | Nigeria | HCW | 422 | Simple random | Mar, 2021 | 32.9 | 30-39(38.6) | 231 |
| 5 | Pharm et al [24] | 2022 | Ethiopia | HCW | 319 | Stratified random | Jun-Jul, 2021 | 53.6 | 20-29(56.7) | 87 |
| 6 | Nnaemeka et al [25] | 2022 | Nigeria | HCW | 710 | Convenience | Sept, 2021-Mar 2022 | 39.6 | 15-25(43.4) | 287 |
| 7 | Nzaji et al [26] | 2020 | DRC | HCW | 613 | Convenience | Mar-Apr, 2020 | 50.9 | 40.3 | 443 |
| 8 | Andrew et al [27] | 2021 | Uganda | MS | 600 | Convenience | Not reported | 62.8 | <24(61.2) | 184 |
| 9 | Eveline et al [28] | 2022 | Tanzania | HCW | 811 | Convenience | Sept 2021 | 52.0 | 35 | 517 |
| 10 | Mohammed et al [29] | 2021 | Ethiopia | HCW | 614 | Convenience | Mar-Jul, 2021 | 48.5 | <30(57.0) | 370 |
| 11 | Mustapha et al [30] | 2021 | Nigeria | MS | 440 | Convenience | Mar-Jun, 2021 | 49.1 | 23 | 264 |
| 12 | Dufera et al [31] | 2021 | Ethiopia | HCW | 522 | Snow ball | Jun 2021 | 90.2 | 30-39(77.8) | 198 |
| 13 | Tolossa et al [32] | 2022 | Ethiopia | HCW | 439 | Proportionate | Apr 2021 | 62.18 | <30(63.6) | 191 |
| 14 | Yendewa et al [33] | 2022 | Sierra Leone | HCW | 592 | Convenience | Jan-Mar, 2022 | 67.2 | 29 | 356 |
| 15 | Botwe et al [34] | 2022 | Ghana | HCW | 108 | Not reported | Feb 2021 | 67.6 | 20-29(50.9) | 44 |

HCW = Healthcare workers; MS = Medical students; CP = Clinical practitioners; HP = Health professionals

concern. The results of the study showed that the pooled prevalence of COVID-19 vaccination hesitancy among healthcare workers in sub-Saharan Africa was 46%. In this meta-analysis, a random effects model was performed as high heterogeneity ($I^2$ = 91.96%, p≤0.001) was detected within the included studies. In particular, despite availability and accessibility, a significant proportion of the healthcare workers in our study refused vaccination, demonstrating that healthcare workers are not immune to vaccine hesitancy. Indeed, the WHO has identified vaccination hesitancy as one of the ten most serious threats to world health [35]. Healthcare workers must be vaccinated against COVID-19 because they give care to COVID-19 patients, and a high infection rate among HCWs could result in a major loss in this critical workforce. Furthermore, healthcare workers who have been vaccinated and are well-informed are a valuable source of COVID-19 vaccine knowledge and are more likely to advocate COVID-19 vaccination to their families, colleagues, and patients [36]. Healthcare workers are respected experts who can act as role models for healthy behavior in the wider public, hence increasing vaccination coverage [37]. This result was within the range of 12 to 91.4% observed in the United States of America [38]. The result was however, higher than the 26.7% reported by a systematic review conducted in the United States and the United Kingdom [39] and 27.8% in Europe [40] and 12.06% and19.15% hesitancy rate in three national studies in China with more than 2,000 participants [41, 42]. This difference may be attributed to differences in sample size.

Sub-group analysis was performed utilizing African areas in this systematic review and meta-analysis because of the presence of significant heterogeneity in the included papers, which may expose the findings to publication bias. The pooled estimated COVID-19 hesitancy rate in Central Africa was 72% (95% CI: 64%–80%), in West Africa it was 52% (CI: 43%–61%), and in East Africa it was 40% (95% CI: 30%–50%). The presence of variability may be attributed to study sample size, study design characteristics, and study settings.

In this systematic and meta-analytical analysis, the most common factors that predicted vaccine hesitancy were negative beliefs towards vaccines, a perceived low risk of COVID-19 infection, and vaccine side effects. It's significant to note that these predictors were also named

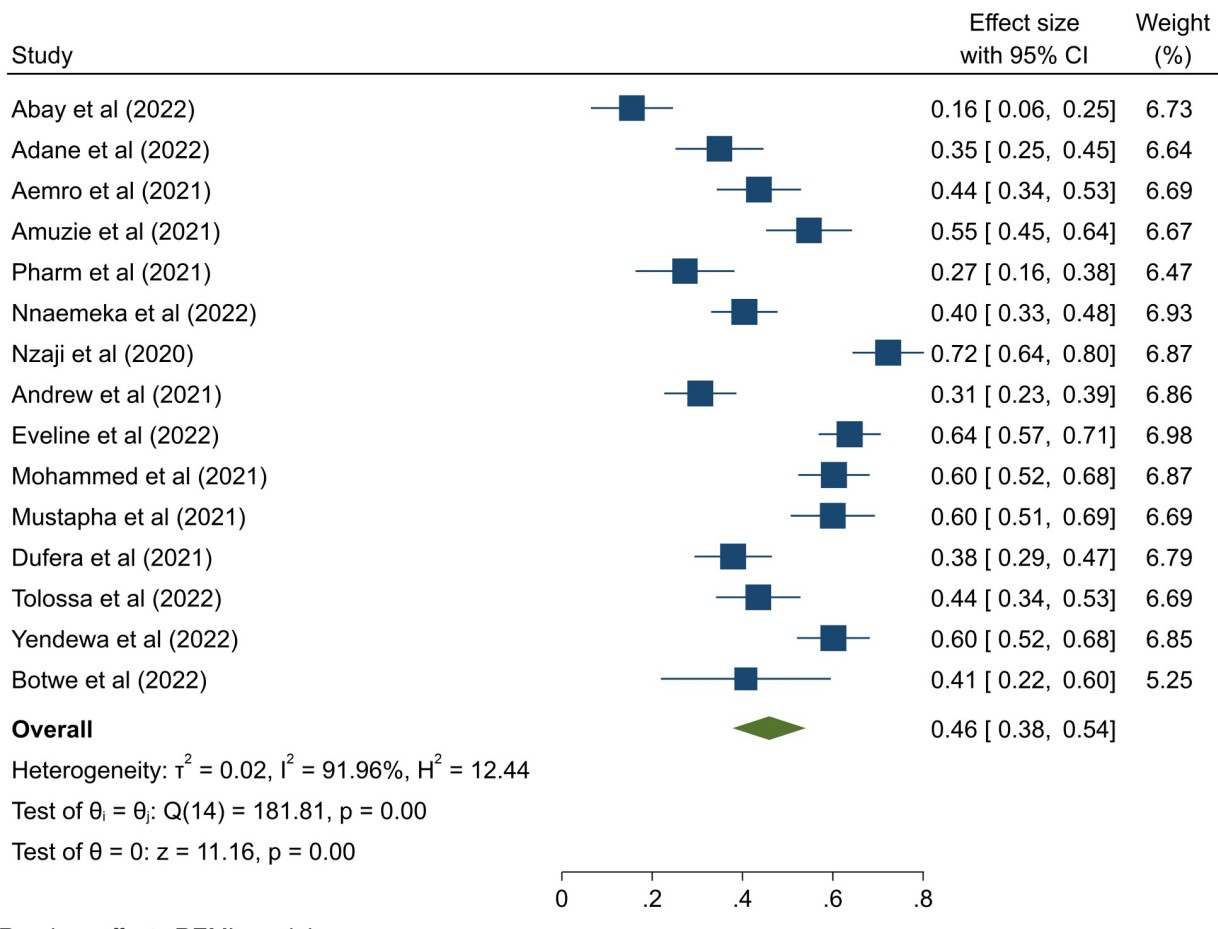

**Fig 2. Forest plot of the pooled proportion of COVID-19 vaccine hesitancy in Sub-Saharan Africa, 2022.**

as pertinent variables in two earlier, related systematic reviews and meta-analyses [43, 44]. Results indicate that healthcare workers who thought that COVID-19 vaccines had side effects were 1.23 times more likely to be hesitant as compared to their counterparts who thought otherwise. Our finding confirms findings from studies done in the United States, [45] China [46] and DRC [26] that participants' main worries and vaccine hesitancy were long- and short-term side effects. Therefore, to increase vaccine uptake and acceptance, the ministry of health should develop strategies such as organizing intercultural health advocacy sessions for healthcare workers and the community, improving health advocates' knowledge and interpersonal communication skills, involving influential community leaders in the dissemination of information, and involving vaccine users in providing agreed vaccination in a variety of cultural contexts [29].

Another important predictor of vaccine hesitancy among healthcare professionals in Sub-Saharan Africa in our study was a lower perceived risk of getting infected. Results also show that healthcare workers perceiving a low risk of contracting COVID-19 were 1.25 times more likely to hesitate the COVID-19 vaccine compared to their counterparts perceiving a high risk of infection. It is significant to note that this predictor was also recognized as a pertinent component in earlier related systematic reviews and meta-analyses [43, 44]. Therefore, this study suggests that improved enactment is required to address issues of confidence, acceptance, benefit, and worry regarding the side effects of the COVID-19 vaccine.

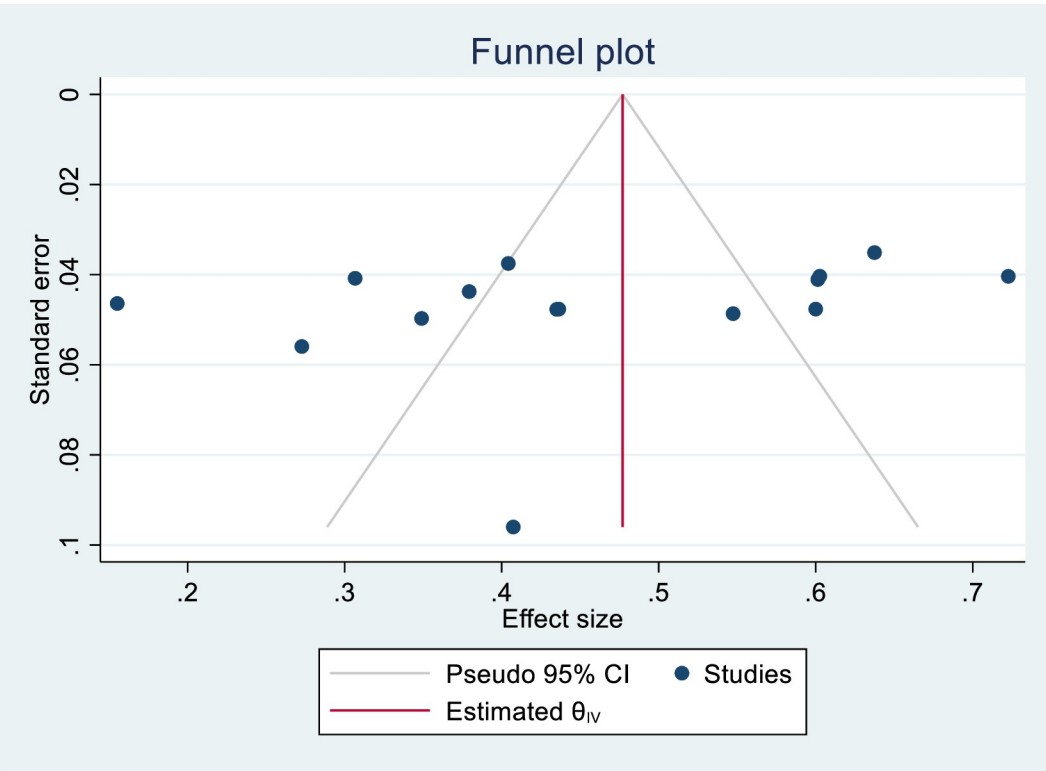

**Fig 3. Assessment of publication bias.**

Our results indicated that healthcare workers who reported negative attitudes and beliefs about the vaccine had a 5% increased likelihood of hesitating vaccination compared to those who did not report negative beliefs. The participants' erroneous beliefs, discomfort, and skepticism may cause hesitation in receiving the COVID-19 vaccine. There is evidence to show that acceptance of the COVID-19 vaccination is significantly influenced by a positive attitude toward the vaccine [47]. Participants who had a positive attitude toward the COVID-19 vaccine were more likely to accept it than those who had a negative attitude [47]. This suggests that having a good attitude about getting vaccinated against COVID-19 is essential. This indicator of COVID-19 vaccination hesitancy discovered here is likely explained by widespread COVID-19 disinformation, such as fake news or the dissemination of inaccurate or false information regarding COVID-19 on social media, such as Facebook or other networks [48, 49].

This review seems to be the first of its kind to synthesize evidence on the prevalence and predictors of COVID-19 vaccination hesitancy among healthcare workers in Sub-Saharan Africa. However, it has the following limitations: Only English-language articles were included in the review. Only quantitative studies were included. Also, we included only primary studies. The systematic review also reports publication bias from small study differences. This might have resulted in omissions or restricted the applicability of our findings.

## Conclusion

The data revealed generally high hesitancy of COVID-19 vaccine among health workers in Sub-Saharan Africa. The following were the most common predictors of COVID-19 vaccine hesitancy: negative beliefs towards vaccines, perceived low risk of COVID-19 infection, and vaccine side effects. Future COVID-19 adoption and uptake should be improved by national

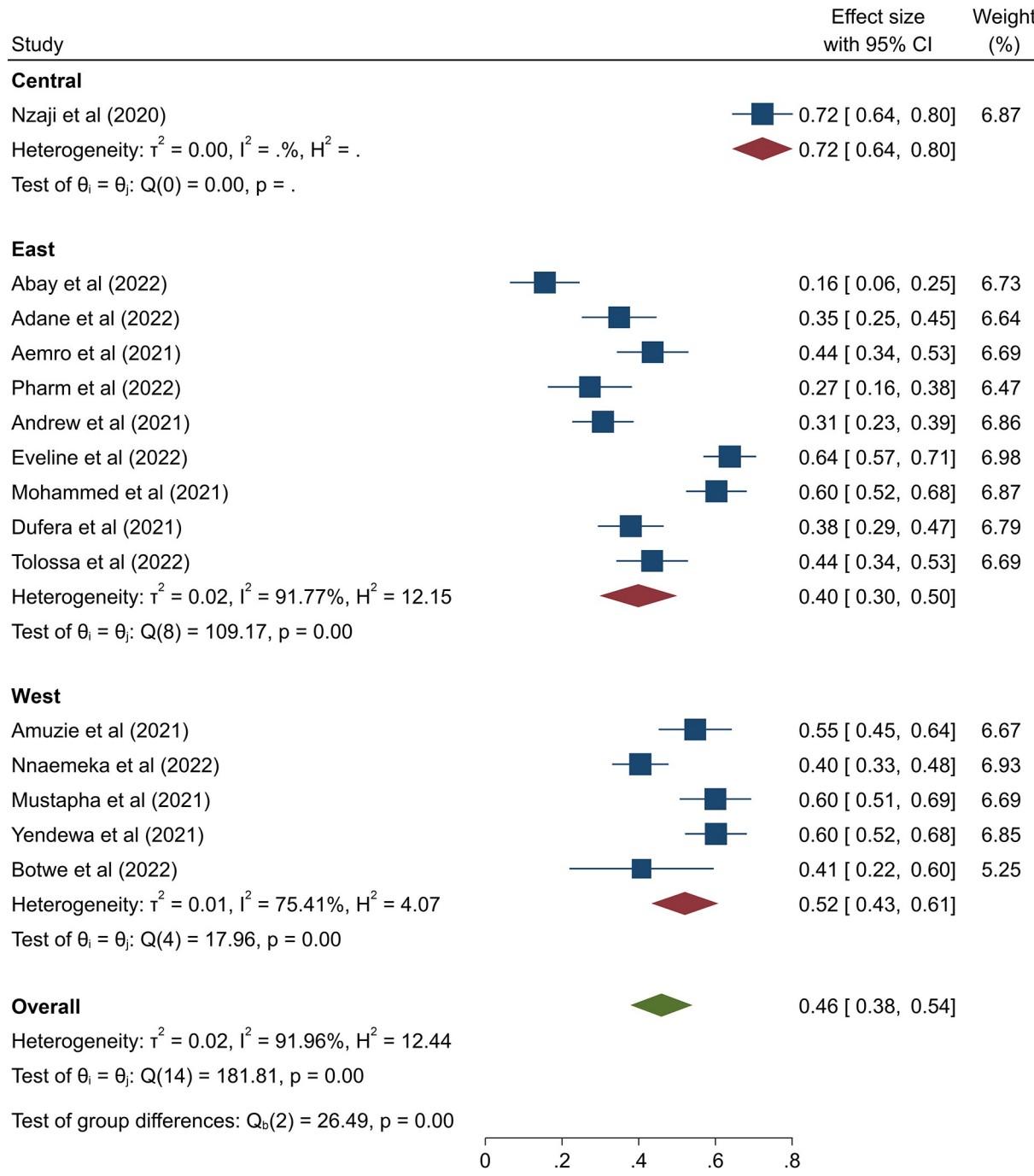

**Fig 4. Forest plot of the pooled proportion of COVID-19 vaccine hesitancy by regions of Sub-Saharan Africa, 2022.**

and individual-level efforts. In Sub-Saharan Africa, it is crucial to address the myths and obstacles preventing healthcare professionals from accepting the COVID-19 vaccination as soon as feasible since their willingness to get the vaccine serves as an important example for the broader public.

**Table 2. Predictors of COVID-19 vaccine hesitancy among health workers in Sub-Saharan Africa.**

| Variable | Studies | Proportion | OR (95% CI) | $I^2$, P-value |
|---|---|---|---|---|
| Negative beliefs towards vaccine | 6 | 14.0 | 1.05(1.04, 1.06) | 98.7%, <0.001 |
| Perceived low risk of COVID-19 infection | 4 | 24.0 | 1.25(1.23, 1.28) | 98.2%, <0.001 |
| Vaccine side effects | 8 | 25.0 | 1.23(1.21, 1.24) | 99.2%, <0.001 |

## Supporting information

**S1 Checklist. PRISMA 2020 checklist.**
(DOCX)

**S1 Table. Search strategy for the databases.**
(DOCX)

**S2 Table. Quality assessment checklist.**
(DOCX)

**S1 Raw data.**
(XLSX)

**S1 Fig. Leave one out sensitivity analysis.**
(DOCX)

## Acknowledgments

The authors acknowledge the authors of the different study articles that were used in the present systematic review and meta-analysis.

## Author Contributions

**Conceptualization:** Eustes Kigongo, Amir Kabunga, Raymond Tumwesigye, Walter Acup.

**Data curation:** Eustes Kigongo, Amir Kabunga, Marvin Musinguzi.

**Formal analysis:** Eustes Kigongo, Amir Kabunga, Walter Acup.

**Funding acquisition:** Ronald Izaruku.

**Investigation:** Eustes Kigongo, Raymond Tumwesigye, Marvin Musinguzi, Walter Acup.

**Methodology:** Eustes Kigongo, Amir Kabunga, Walter Acup.

**Resources:** Marvin Musinguzi, Ronald Izaruku.

**Software:** Raymond Tumwesigye, Marvin Musinguzi, Ronald Izaruku.

**Supervision:** Amir Kabunga, Raymond Tumwesigye, Marvin Musinguzi, Ronald Izaruku.

**Validation:** Raymond Tumwesigye, Marvin Musinguzi, Ronald Izaruku.

**Visualization:** Raymond Tumwesigye, Marvin Musinguzi, Ronald Izaruku, Walter Acup.

**Writing – original draft:** Eustes Kigongo, Amir Kabunga, Raymond Tumwesigye, Marvin Musinguzi, Ronald Izaruku, Walter Acup.

**Writing – review & editing:** Eustes Kigongo, Amir Kabunga, Walter Acup.

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
