## [Decision Letter · Decision Letter 0]

25 Apr 2023

PONE-D-22-29316Prevalence and Predictors of COVID-19 vaccination hesitance among healthcare workers in Sub-Saharan Africa: a systematic review and meta-analysisPLOS ONE

Dear Dr. Kigongo,

Thank you for submitting your manuscript to PLOS ONE. After careful consideration, we feel that it has merit but does not fully meet PLOS ONE’s publication criteria as it currently stands. Therefore, we invite you to submit a revised version of the manuscript that addresses the points raised during the review process.

The reviewers have provided feedback on the manuscript to enhance its quality. The authors are advised to consider the following key points for any subsequent revisions:

Carefully review the manuscript for language errors and correct them as needed.Use the PRISMA 2020 flow diagram and cite PRISMA (http://www.prisma-statement.org/PRISMAStatement/CitingAndUsingPRISMA.aspx).Improve the introduction by providing more justification for the research questions.Consider assessing the impact of heterogeneity and bias on the reported results.Ensure consistency among calculations, text, figures, and tables in the results section.To improve reproducibility, ensure that supporting data are available in the manuscript’s supplementary files or public repositories. This can include a detailed search strategy used for each database and a detailed assessment of risk of bias for each included study.Include a discussion of study limitations in the manuscript.

We look forward to receiving your revised manuscript.

Kind regards,

Delfina Fernandes Hlashwayo, M.Sc.

Academic Editor

PLOS ONE

Journal Requirements:

 “no response”

“no response”

Reviewers' comments:

Reviewer's Responses to Questions

**Comments to the Author**

1. Is the manuscript technically sound, and do the data support the conclusions?

Reviewer #1: Yes

Reviewer #2: Yes

Reviewer #3: Yes

2. Has the statistical analysis been performed appropriately and rigorously? 

Reviewer #1: No

Reviewer #2: Yes

Reviewer #3: Yes

3. Have the authors made all data underlying the findings in their manuscript fully available?

Reviewer #1: Yes

Reviewer #2: Yes

Reviewer #3: Yes

4. Is the manuscript presented in an intelligible fashion and written in standard English?

Reviewer #1: No

Reviewer #2: Yes

Reviewer #3: Yes

5. Review Comments to the Author

Reviewer #1: Topic : Prevalence and Predictors of COVID-19 vaccination hesitance among healthcare workers in Sub-Saharan Africa: a systematic review and meta-analysis

Manuscript ID: PONE-D-22-29316

The systematic review and meta-analysis done by Kigongo etin prevalence and predictors of COVID-19 vaccination hesitance has a public health significance and a timely topic; however there are a lot of issues and must be corrected before publication.

Minor comments

1. In abstract correct this sentence. Data were extracted using was extracted using excel.

2. Topographical errors like grammar and spelling correct it.

Major comments

1. Only cross sectional studies included what about case control, and cohort studies?

2. What about studies written other than English language?

3. All the studies regardless of the level of risk were included. So what is the significance of quality assessment, if you included all studies without assessing risk of bias. It is better to avoid those studies high risk of bias or weak studies and reanalyze?

4. In result part in Figure 1, you have excluded 11 studies from 35, you should included 24 studies but you have included only 22 studies ? Correct and reanalyze

5. Under characteristics of studies you have mentioned 23 studies but you have included 22studies? Please correct it.

6. There is a table in page ten without any table name 2 or 3 ?

7. The pooled prevalence of vaccine hesitance was 36% and I2 =99.58% , this shows heterogeneity is very high. If heterogeneity is high, it is difficult to pool. If you want to pool the findings you should do other heterogeneity assessment methods like sub group analysis or using meta regression, but I did not see such methods here?

8. In this manuscript there is no strength and limitation of this systematic and meta-analysis?

Reviewer #2: Abstract

The abstract speaks to the systematic review. The problem is stated with relevant support the systematic review. The methods shows summary of the articles used but were limited to cross-sectional studies of a period of 2 years (2020-2022). This is reasonable period since the COVID-19 pandemic started in 2019 achieving appropriate results. The abstract concludes with summary of the findings relevant to the subject. The results are pertinent to the local region

Background

1. The review was not of randomized trials. The review included the description of the approaches used to identify all the literature in the subject. The question the paper wants to answer should be stated clearly. Moreover, it is surprising that the review could not capture the study below which was conducted in a West African country among a specific healthcare professionals about COVID -19 hesitancy in vaccine uptake to give more credence to the systematic study. It is advised the authors include this paper in the review unless they found it in the eleven articles removed as stated in the results.

Botwe BO, Antwi WK, Adusei JA, Mayeden RN, Akudjedu TN, Sule SD. COVID-19 vaccine hesitancy concerns: Findings from a Ghana clinical radiography workforce survey. Radiography (Lond). 2022 May;28(2):537-544. Doi: 10.1016/j.radi.2021.09.015. Epub 2021 Oct 8. PMID: 34654631; PMCID: PMC8498685.

2. The context of the systematic review including why it’s important are stated.

3. The statement in page (8-9) “COVID-19 vaccine hesitance and acceptance among Healthcare workers in Sub-Saharan Africa have been investigated through observational research [4]” cannot be totally true as the study quoted above was not an observational study.

Why Methods and methods?

Looking at the period of the unset of the pandemic the participants identified in the 22 articles and the aggregate data used is large enough for the systematic review.

Search Engine

Appropriate but would be good to include SCOPUS articles if possible.

Eligibility criteria

Inclusion

The review included only studies reporting on predictors of COVID-19 vaccination hesitance. However, Botwe et al study indicated above reported similar findings

Results

Data was synthesized.

1. The writers offer a high-quality, well-protected setting for assessing significant research works that advance understanding of the subject matter and a specific geographic area.

2. The study’s characteristics are displayed clearly in the results and supported with Forest plot of the pooled proportion of COVID-19 vaccine hesitance.

3. However, the Table with the heading “Predictors of COVID-19 vaccine hesitance among Health workers in Sub-Saharan Africa” is supposed to come with a label (i.e Table 2)

4. Overall the results will help local health policies

Discussion.

The discussion was tailored to the results and in relation to previous studies used for the systematic review.

Reviewer #3: Please find my specific review comments below:

Background:

(1) The pooled COVID-19 vaccine hesitancy in Africa was 54%[4]. It is difficult to

comprehend vaccination hesitancy since it is influenced by complicated and context-specific elements that change

across time, place, and vaccines. Complacency, convenience, confidence, as well as sociodemographic and cultural

factors also have an impact on the outcome[3].

Here, are the context specific elements and sociodemographic and cultural factors different? How?

(2) A safe and effective vaccine is the most effective and dependable way to build up the population's immune system to prevent recurrent infections[5].

Here, is population's immune system refering to herd immunity? please clarify

(3) The efficiency of a vaccine and the extent to which it is used determines its success[9]

Please elaborate the 'extent' and please make more logical connection with the next sentences.

(4) However, among this group of people in the region, there are currently no thorough reviews.

Please strengthen the justification part. Now this is not much stronger.

(5) any reviews and experimental studies were excluded.

Please provide justification

(6) qualitative studies were excluded

Please provide reasons.

(7) After screening abstracts and titles, 35 articles remained and were subjected to full-text screening.

Please elaborate the screening process here.

(8) the wrong publication type (n=3)

What is meant by wrong publication type?

(9) Table 1: One collumn for sampling techniques can be added

(10) The result implies that a considerable proportion of healthcare workers in Sub-Saharan Africa were hesitant towards the COVID-19 vaccine, implying a direct negative impact on the mitigation of COVID-19.

Hesitant towards receiving? Please make it clear here.

(11) the concerned bodies like ministries of health should develop strategies such as ........

what about other concerned bodies? who are they? please mention

6. PLOS authors have the option to publish the peer review history of their article (what does this mean?). If published, this will include your full peer review and any attached files.

Reviewer #1: No

Reviewer #2: **Yes: **DR. WILLIAM KWADWO ANTWI

Reviewer #3: **Yes: **Shafayat Sultan

---

## [Author Response · Author response to Decision Letter 0]

15 Jun 2023

Eustes Kigongo

Department of Environmental Health and Disease Control, Lira University

Uganda 

May 2, 2023

Dear Editor,

Re: Response to reviewers’ comments 

First all, I wish to thank the reviewers and editors for the positive feedback that will help to improve the quality of our manuscript. 

This letter serves to re-submit our manuscript titled “Prevalence and Predictors of COVID-19 vaccination hesitance among healthcare workers in Sub-Saharan Africa: a systematic review and meta-analysis”. 

Please see the responses in the table below;

Sn Comments Response Page, Lines

 Editor 

1 Carefully review the manuscript for language errors and correct them as needed. Thanks for this

The manuscript has been reviewed All pages 

2 Use the PRISMA 2020 flow diagram and cite PRISMA (http://www.prisma-statement.org/PRISMAStatement/CitingAndUsingPRISMA.aspx)

Thank you,

This has been used Pg 6,

Lines 166-200

3 Improve the introduction by providing more justification for the research questions Thanks for this advice 

The introduction has been improved by providing more justification for the research questions Pg 3, 

Lines 76-87

4 Consider assessing the impact of heterogeneity and bias on the reported results Thank you for the insight,

Heterogeneity and bias has been assessed using Egger’s test and a funnel plot inserted as supplementary 1 Pg 5,

Lines 143-150

5 Ensure consistency among calculations, text, figures, and tables in the results section Thank you, these have been updated throughout the entire manuscript All pages

6 To improve reproducibility, ensure that supporting data are available in the manuscript’s supplementary files or public repositories. This can include a detailed search strategy used for each database and a detailed assessment of risk of bias for each included study Thank you for the comment,

The detailed search strategy has been included as a supplementary 2

The risk of bias for each included study was assessed and the tool used has been cited. Pg 4, 

Lines 103-104

Pg 4,

Lines 126-127

7 Include a discussion of study limitations in the manuscript We have included the limitations in the manuscript Pg 13,

Lines 288-292

 Reviewer one 

1 In abstract correct this sentence. Data were extracted using was extracted using excel Thanks for this

This has been rectified Pg 1,

Lines 13

2 Topographical errors like grammar and spelling correct it Thanks for this

The manuscript has been reviewed All pages

3 Only cross sectional studies included what about case control, and cohort studies? Thank you,

This was a typo and has been rectified. Pg 4,

Lines 107-110

4 What about studies written other than English language? Thanks for this observation, ‘

We included on studies in English. However, we have acknowledged this in the limitations of the study. Pg 13,

Lines 288-292

5 All the studies regardless of the level of risk were included. So what is the significance of quality assessment, if you included all studies without assessing risk of bias. It is better to avoid those studies high risk of bias or weak studies and reanalyze? Thank you for the observation,

This was a typological error, only studies with low and moderate risk were included. Pg 5,

Lines 130-131

6 In result part in Figure 1, you have excluded 11 studies from 35, you should included 24 studies but you have included only 22 studies? Correct and reanalyze Thank you, 

This has been rectified Pg 5,

Lines 153-155

7 Under characteristics of studies you have mentioned 23 studies but you have included 22studies? Please correct it. Thank you, 

This has been rectified Pg 7,

Lines 203-206

 Reviewer Two 

1 The abstract speaks to the systematic review. The problem is stated with relevant support the systematic review. The methods shows summary of the articles used but were limited to cross-sectional studies of a period of 2 years (2020-2022). This is reasonable period since the COVID-19 pandemic started in 2019 achieving appropriate results. The abstract concludes with summary of the findings relevant to the subject. The results are pertinent to the local region Thanks for the complement NA

2 The review was not of randomized trials. The review included the description of the approaches used to identify all the literature in the subject. The question the paper wants to answer should be stated clearly. Thank you for the great suggestion,

The questions have been clearly stated Pg 3,

Lines 88-91

 Moreover, it is surprising that the review could not capture the study below which was conducted in a West African country among a specific healthcare professionals about COVID -19 hesitancy in vaccine uptake to give more credence to the systematic study. It is advised the authors include this paper in the review unless they found it in the eleven articles removed as stated in the results Thank you,

This study has been considered Pg 8,

Lines 212-213 

4 The context of the systematic review including why it’s important are stated Thanks for the complement NA

5 The statement in page (8-9) “COVID-19 vaccine hesitance and acceptance among Healthcare workers in Sub-Saharan Africa have been investigated through observational research [4]” cannot be totally true as the study quoted above was not an observational study Thanks for this observation 

This statement has been modified Pg 3,

Lines 76-87

6 Looking at the period of the unset of the pandemic the participants identified in the number of articles and the aggregate data used is large enough for the systematic review. Thank you for the compliment NA

7 The review included only studies reporting on predictors of COVID-19 vaccination hesitance. However, Botwe et al study indicated above reported similar findings Thank you,

This study has been incorporated Pg 8,

Lines 212-213

8 Data was synthesized.

The writers offer a high-quality, well-protected setting for assessing significant research works that advance understanding of the subject matter and a specific geographic area Thanks for this complement NA

9 The study’s characteristics are displayed clearly in the results and supported with Forest plot of the pooled proportion of COVID-19 vaccine hesitance Thanks for this complement NA

10 However, the Table with the heading “Predictors of COVID-19 vaccine hesitance among Health workers in Sub-Saharan Africa” is supposed to come with a label (i.e Table 2) Thank you,

This has been done Pg 11,

Lines 232

11 Overall the results will help local health policies Thanks NA

12 Discussion.

The discussion was tailored to the results and in relation to previous studies used for the systematic review. Thanks for this complement NA

 Reviewer Three 

1 The pooled COVID-19 vaccine hesitancy in Africa was 54%[4]. It is difficult to comprehend vaccination hesitancy since it is influenced by complicated and context-specific elements that change across time, place, and vaccines. Complacency, convenience, confidence, as well as sociodemographic and cultural factors also have an impact on the outcome[3]. Thank you for the observation,

The statement is rectified Pg 1 & 2,

Lines 39-41

3 A safe and effective vaccine is the most effective and dependable way to build up the population's immune system to prevent recurrent infections [5].

Here, is population's immune system referring to herd immunity? please clarify Thanks for this

We clarified this in the text;

'population immunity is the same as herd immunity' is Pg 3,

Lines 76-87

4 The efficiency of a vaccine and the extent to which it is used determines its success[9]

Please elaborate the 'extent' and please make more logical connection with the next sentences. Thanks for this information 

The statement has been modified Pg 3,

Lines 76-87

5 However, among this group of people in the region, there are currently no thorough reviews.

Please strengthen the justification part. Now this is not much stronger. the introduction has been improved by providing more justification for the research questions Pg 3,

Lines 76-87

6 Any reviews and experimental studies were excluded.

Please provide justification It's true, we only included only primary studies. However, we have acknowledged this in the limitations of the study.

Secondly at the time of the review experimental studies seemed minimal or non-existent Pg 13,

Lines 288-292

7 Qualitative studies were excluded

Please provide reasons. Thank you,

This has been acknowledged as a limitation Pg 13,

Lines 288-292

8 After screening abstracts and titles, 35 articles remained and were subjected to full-text screening.

Please elaborate the screening process here. Thank you for the insight,

The screening process is elaborated in the study and data management section Pg 4,

Lines 113-119

9 The wrong publication type (n=3)

What is meant by wrong publication type? Thank you for the good observation,

This has been re-written Pg 5,

Lines 153-155

10 Table 1: One column for sampling techniques can be added Thank you for the advice,

This has been added Pg 8,

Lines 211-213

11 The result implies that a considerable proportion of healthcare workers in Sub-Saharan Africa were hesitant towards the COVID-19 vaccine, implying a direct negative impact on the mitigation of COVID-19.

Hesitant towards receiving? Please make it clear here. Thanks for this advice

We have modified the sentences Pg 11,

Lines 283-241

12 The concerned bodies like ministries of health should develop strategies such as ........

What about other concerned bodies? who are they? please mention Thanks for this

We have modified the sentence Pg 12,

Lines 265

---

## [Decision Letter · Decision Letter 1]

11 Jul 2023

PONE-D-22-29316R1Prevalence and Predictors of COVID-19 vaccination hesitance among healthcare workers in Sub-Saharan Africa: a systematic review and meta-analysisPLOS ONE

Dear Dr. Kigongo,

Thank you for submitting your manuscript to PLOS ONE. After careful consideration, we feel that it has merit but does not fully meet PLOS ONE’s publication criteria as it currently stands. Therefore, we invite you to submit a revised version of the manuscript that addresses the points raised during the review process.

The reviewers have provided feedback on the manuscript to enhance its quality. The authors are advised to consider the following key points for any subsequent revisions:

Abstract: Please ensure that the results are not repeated in the conclusion. The conclusion should provide a concise summary of the findings and highlight their implications and significance.Please review the order of keywords to ensure consistency and alignment with relevant concepts in the manuscript.Ensure consistent use of terms related to vaccine hesitancy throughout the manuscript to maintain clarity and coherence.Please provide a comprehensive and well-defined set of exclusion criteria that clearly outline the criteria for excluding studies from the systematic review.Please clarify if the tool adapted from Hoy et al. has been validated. If any item was removed, provide a justification for its exclusion. In addition, please include the questions and scoring criteria for the risk of bias assessment in the supplementary file, as requested previously.Please use the correct symbol (p<0.05) to indicate statistical significance, rather than using "P=0.05."Include the search strategies for Google Scholar, African Journal Online, and Science Direct in the supplementary file, along with the date and results in numbers. Currently, the supplementary file only includes the search strategy for PubMed.Please provide full names for the abbreviations used in the Supplementary File 2 and ensure that the citations contain the corresponding references.Please revise the first sentence of the conclusion to accurately reflect the quantitative data and align it with the findings of the study. Please carefully review the comments provided by Reviewer 4 in the attached PDF and address them accordingly. Please submit your revised manuscript by Aug 25 2023 11:59PM. If you will need more time than this to complete your revisions, please reply to this message or contact the journal office at plosone@plos.org. Please include the following items when submitting your revised manuscript:A rebuttal letter that responds to each point raised by the academic editor and reviewer(s). You should upload this letter as a separate file labeled 'Response to Reviewers'.A marked-up copy of your manuscript that highlights changes made to the original version. You should upload this as a separate file labeled 'Revised Manuscript with Track Changes'.An unmarked version of your revised paper without tracked changes. You should upload this as a separate file labeled 'Manuscript'.If applicable, we recommend that you deposit your laboratory protocols in protocols.io to enhance the reproducibility of your results. Protocols.io assigns your protocol its own identifier (DOI) so that it can be cited independently in the future. For instructions see: https://journals.plos.org/plosone/s/submission-guidelines#loc-laboratory-protocols. Additionally, PLOS ONE offers an option for publishing peer-reviewed Lab Protocol articles, which describe protocols hosted on protocols.io. Read more information on sharing protocols at https://plos.org/protocols?utm_medium=editorial-email&utm_source=authorletters&utm_campaign=protocols.

We look forward to receiving your revised manuscript.

Kind regards,

Delfina Fernandes Hlashwayo, M.Sc.

Academic Editor

PLOS ONE

Journal Requirements:

Reviewers' comments:

Reviewer's Responses to Questions

**Comments to the Author**

1. If the authors have adequately addressed your comments raised in a previous round of review and you feel that this manuscript is now acceptable for publication, you may indicate that here to bypass the “Comments to the Author” section, enter your conflict of interest statement in the “Confidential to Editor” section, and submit your "Accept" recommendation.

Reviewer #2: All comments have been addressed

Reviewer #4: (No Response)

2. Is the manuscript technically sound, and do the data support the conclusions?

Reviewer #2: Yes

Reviewer #4: Yes

3. Has the statistical analysis been performed appropriately and rigorously? 

Reviewer #2: N/A

Reviewer #4: Yes

4. Have the authors made all data underlying the findings in their manuscript fully available?

Reviewer #2: (No Response)

Reviewer #4: Yes

5. Is the manuscript presented in an intelligible fashion and written in standard English?

Reviewer #2: Yes

Reviewer #4: Yes

6. Review Comments to the Author

Reviewer #2: Issues raised in the previous review have been addressed. What I cannot find per the manuscript sent to me is the data availability. It only contains the response to reviewer comments. Have you made all data underlying the findings in the manuscript fully available?

Reviewer #4: (No Response)

7. PLOS authors have the option to publish the peer review history of their article (what does this mean?). If published, this will include your full peer review and any attached files.

Reviewer #2: **Yes: **Dr William Kwadwo Antwi

Reviewer #4: **Yes: **Chinedu Anthony Iwu

---

## [Author Response · Author response to Decision Letter 1]

14 Jul 2023

Eustes Kigongo

Department of Environmental Health and Disease Control, Lira University

Uganda 

July 14, 2023

Dear Editor,

Re: Response to reviewers’ comments 

First all, I wish to thank the reviewers and editors for the positive feedback that will help to improve the quality of our manuscript. 

This letter serves to re-submit our manuscript titled “Prevalence and Predictors of COVID-19 vaccination hesitance among healthcare workers in Sub-Saharan Africa: a systematic review and meta-analysis”. 

Please see the responses in the table below;

SN Reviewers’ comments Response Page/lines

1 Abstract: Please ensure that the results are not repeated in the conclusion. The conclusion should provide a concise summary of the findings and highlight their implications and significance Thank you,

We have removed the results from the conclusion. Pg 1,

Lines 35-36

2 Please review the order of keywords to ensure consistency and alignment with relevant concepts in the manuscript. Thank you,

The key words have been reorganized as advised. Pg 1,

Lines 40

3 Ensure consistent use of terms related to vaccine hesitancy throughout the manuscript to maintain clarity and coherence. For consistency, we have used “Hesitancy” throughout the entire manuscript. All pages

4 Please provide a comprehensive and well-defined set of exclusion criteria that clearly outline the criteria for excluding studies from the systematic review. Thank you,

We have provided the exclusion criteria. Pg 4,

Lines 123-125

5 Please clarify if the tool adapted from Hoy et al. has been validated. If any item was removed, provide a justification for its exclusion. In addition, please include the questions and scoring criteria for the risk of bias assessment in the supplementary file, as requested previously. This has been clarified, we used the validated tool and we did not modify.

The tool has been attached as supplementary file 3. Pg 5,

Lines 139-140

6 Please use the correct symbol (p<0.05) to indicate statistical significance, rather than using "P=0.05." Thank you,

This has been rectified. Pg 5,

Lines 158

7 Include the search strategies for Google Scholar, African Journal Online, and Science Direct in the supplementary file, along with the date and results in numbers. Currently, the supplementary file only includes the search strategy for PubMed. Thank you,

This has been addressed accordingly. Attached as Supplementary file 1 NA

8 Please provide full names for the abbreviations used in the Supplementary File 2 and ensure that the citations contain the corresponding references. Thank you,

This has been done. NA

9 Please revise the first sentence of the conclusion to accurately reflect the quantitative data and align it with the findings of the study. Thank you,

This has been revised. Pg 14,

Lines 311-312

10 Please carefully review the comments provided by Reviewer 4 in the attached PDF and address them accordingly. Thank you for the comments,

Comments in the attached document (manuscript) have been reviewed. All document

---

## [Editor Report · Decision Letter 2]

17 Jul 2023

Prevalence and Predictors of COVID-19 vaccination hesitance among healthcare workers in Sub-Saharan Africa: a systematic review and meta-analysis

PONE-D-22-29316R2

Dear Dr. Kigongo,

We’re pleased to inform you that your manuscript has been judged scientifically suitable for publication and will be formally accepted for publication once it meets all outstanding technical requirements.

Kind regards,

Delfina Fernandes Hlashwayo, M.Sc.

Academic Editor

PLOS ONE
---

## [Editor Report · Acceptance letter]

20 Jul 2023

PONE-D-22-29316R2 

Prevalence and Predictors of COVID-19 vaccination hesitancy among healthcare workers in Sub-Saharan Africa: a systematic review and meta-analysis 

Dear Dr. Kigongo:

I'm pleased to inform you that your manuscript has been deemed suitable for publication in PLOS ONE. Congratulations! Your manuscript is now with our production department. 

Kind regards, 

on behalf of

Ms. Delfina Fernandes Hlashwayo 

Academic Editor

PLOS ONE